# PerspectiveNet: A Scene-consistent Image Generator for New View Synthesis in Real Indoor Environments

**David Novotny**     **Benjamin Graham**     **Jeremy Reizenstein**

Facebook AI Research

London

{dnovotny,benjamingraham,reizenstein}@fb.com

## Abstract

Given a set of a reference RGBD views of an indoor environment, and a new viewpoint, our goal is to predict the view from that location. Prior work on new-view generation has predominantly focused on significantly constrained scenarios, typically involving artificially rendered views of isolated CAD models. Here we tackle a much more challenging version of the problem. We devise an approach that exploits known geometric properties of the scene (per-frame camera extrinsics and depth) in order to warp reference views into the new ones. The defects in the generated views are handled by a novel RGBD inpainting network, PerspectiveNet, that is fine-tuned for a given scene in order to obtain images that are geometrically consistent with all the views in the scene camera system. Experiments conducted on the ScanNet and SceneNet datasets reveal performance superior to strong baselines.

## 1   Introduction

Decisions often have to be made on the basis of incomplete information about our visual environment. Humans instinctively fill the gaps in from prior experience. This is an enabler of many tasks such as navigation, and machine learning should strive to match this ability. One way of quantitatively measuring it is via generating new views within a partially explored environment. Many variants of this problem, known as *new view synthesis*, exist, ranging from a category-specific setup, where the hallucinated views are conditioned on image(s) of an isolated instance of a well defined visual object category (car, chair) [9], to inferring new photo-realistic pictures of outdoor or indoor scenes given a set of reference images [13]. The former can be seen as a subtask of the latter, since real scenes contain many instances of various object categories in an arbitrary geometric configuration. Perhaps due to the challenging nature of the more unconstrained setup, the community in recent years mostly focused on the category-specific scenario, restricting a large portion of the experimental evaluation to clean synthetic datasets such as ShapeNet [4].

In this work, we take a step toward the more complex task of generating new views of real indoor environments. Historically, the new view synthesis task has been addressed with either learning based methods [43, 20, 19], that leverage deep nets to map an encoding of a viewpoint and style to a new view, or methods that exploit geometric properties of a given scene to warp reference images into a target viewpoint, usually with some human intervention, possibly followed by an inpainting step that fills the newly appearing holes [17, 7]. While learning based methods are suitable for the category-specific setup, where the viewpoint-to-image mapping is less complex, due to the regular geometric structure of object categories, the indoor scene synthesis was predominantly addressed with different variants of the render-inpaint technique. Similar to previous approaches, we tackle the task by devising a novel variant of the render-inpaint approach.

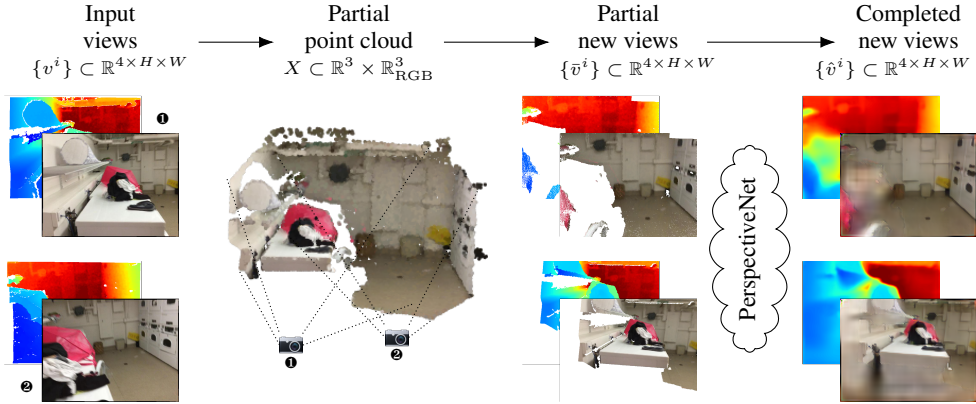

Figure 1: **New view synthesis in real indoor environments.** Given a sparse set of reference RGBD room views, our goal is to generate new views of the same room.

Our main contribution is a novel scene-level multi-camera optimization scheme termed *PerspectiveNet*. The crux of the method lies in regarding the joint set of reference and generated test views as a calibrated optical system. Its known physical properties allow for exploiting powerful constraints that enforce consistency of the newly generated views across all present cameras. Combined with a latent representation of pixels in each view, we optimize over the set of latent image codes to generate globally consistent set of views of a considered room.

We conduct one of the first systematic evaluations of the new view synthesis task in the context of real indoor environments. Our method is compared with a variety of strong baselines that either follow the render-inpaint paradigm, or reason about scene contents in 3D space with 3D convolutions. Evaluation on the ScanNet and SceneNet datasets reveals that our method outperforms all baselines both qualitatively and quantitatively.

## 2 Related Work

**Category-specific new view synthesis**    Neural networks can successfully learn complex mappings, including changes of appearance induced by a camera movement. Therefore they have been the method of choice in tackling new view synthesis. The task was mostly explored for isolated object categories such as chairs or faces, since their regular structure significantly constrains the problem. Given an encoding of a relative transformation and a reference image, Transforming Autoencoder [16] produced its transformed version. While [16] applied the architecture to images of digits, Multiview Perceptron [46] synthesized views of human faces.

With the advent of deep learning, convolutional neural networks (CNNs) enabled new view synthesis for more complex object categories. Dosovitskiy et al. [9] generated new views of chairs from a synthetic dataset (ShapeNet). Following [34], [34, 39, 22] proposed a similar encoder-decoder architecture that, differently from [9], generated views of object instances previously unseen in the train set. Similar to our approach, several other methods proposed an alternative that transfers pixels from the reference views, followed by an image refinement step [31, 43, 19]. Other approaches [20] involve an intermediate 3DCNN that aggregates information from the reference views, followed by a learned 3D-to-image decoder. While the aforementioned methods show impressive results, they are restricted to isolated views of object categories from a synthetic dataset. We differ by considering a much more challenging setup with the reference views coming from real indoor environments captured with a hand-held camera.

**New view synthesis in the wild**    Only very few works explored unconstrained generation of new views in real environments. Flynn et al. [13] consider a simpler version of the task where the reference views cover most of the frustum of the test views allowing to form the majority of the synthesized image by copying pixels from the reference views. Eslami et al. [12] proposed an end-to-end trained Generative Query Network (GQN) that renders new viewpoints given a latent encoding of the scene and a novel viewpoint. GQN can effortlessly browse simple synthetic environments, however it has

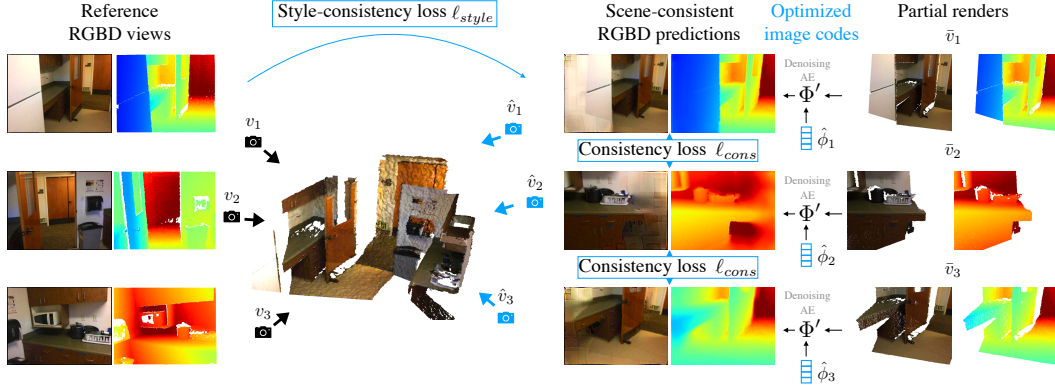

Figure 2: **Scene-consistent optimization.** For a given test scene, PerspectiveNet optimizes the latent representations of new views in order to obtain a scene-consistent set of images that satisfy geometric re-projection constraints of the scene camera system and have similar visual style.

not been tested in a real world setup. The recent method of Meshry et al. [29] captures a complete distribution of possible appearance variations of a, mostly hole-free, image. This differs from our aim of inpainting large undefined regions. Recently, [33] trained a 3D ConvNet for generation of new views of a single non-synthetic object instance.

**Image inpainting** Our method is also related to image inpainting. Early approaches [2, 1, 35, 10, 23] were recently outperformed by deep methods. Isola et al. [18] used a Conditional Generative Adversarial Network cGAN to translate between different types of pixel-wise labels. Many improvements of the original cGAN architecture, including [38, 45, 44], were later proposed. Avoiding the use of GANs, [26] leverage partial convolutions in combination with the perceptual and style losses [14] and achieve state-of-the-art results in semantic image inpainting. Recently, Ulyanov et al. [37] demonstrated that convolutional layers constitute a strong prior for image denoising and, as such, can be used for image denoising without prior training on a dataset of images.

## 3 Method

**Task and naming conventions** Our goal in this paper is to generate new views of an indoor scene given a set of reference views captured by a handheld RGBD camera. More formally, we take as input a set of $N_{\text{ref}}$ reference RGBD views $\{v^i\}_{i=1}^{N_{\text{ref}}}, v^i \in \mathbb{R}^{4 \times H \times W}$ annotated with their corresponding camera extrinsic and intrinsic matrices $g^i \in SE(3)$ and $K^i \in \mathbb{R}^{4 \times 4}$ respectively[1]. At some pixels, the depth value is incorrectly recorded as zero to denote missing data. Given camera parameters $\{(\hat{K}^i, \hat{g}^i)\}_{i=1}^{N_{\text{test}}}$ of $N_{\text{test}}$ test views, our method attempts to generate their RGBD content with a prediction $\{\hat{v}^i\}_{i=1}^{N_{\text{test}}}$. We denote $V = \{v^i\}_{i=1}^{N_{\text{ref}}} \cup \{\hat{v}^i\}_{i=1}^{N_{\text{test}}}$ as a set of all views in a given scene.

Throughout this paper we denote image spatial locations $u = (u_1, u_2) \in \{1, ..., W\} \times \{1, ..., H\}$. At each pixel $u$, we can identify the corresponding per-pixel depth $d_u \in \mathbb{R}$ and color $c_u \in [0, 1]^3$. The knowledge of camera parameters and depth allows to back-project each pixel $u^i = (u_1, u_2)$ from image $i$ to its corresponding 3D point $x_u^i \sim (K^i g^i)^{-1}[u_1, u_2, d_u, 1]^T$ in the common coordinate frame of the corresponding scene. Since we work with rendering algorithms that occasionally produce holes in images (i.e. pixels with undefined color), we denote by $\Omega(v)$ a set of all locations $u$ in a view $v$ that are non-holes (pixels with defined color).

In what follows, we describe a render-inpaint baseline followed by our main contribution consisting of an extension of the baseline to a novel scene-consistent inpainting method, PerspectiveNet.

### 3.1 Inpainting with a denoising RGBD autoencoder

As outlined above, we take a pragmatic approach and start by "copying" all possible pixels from the reference views into the test ones. While this can be achieved with depth-based image rendering

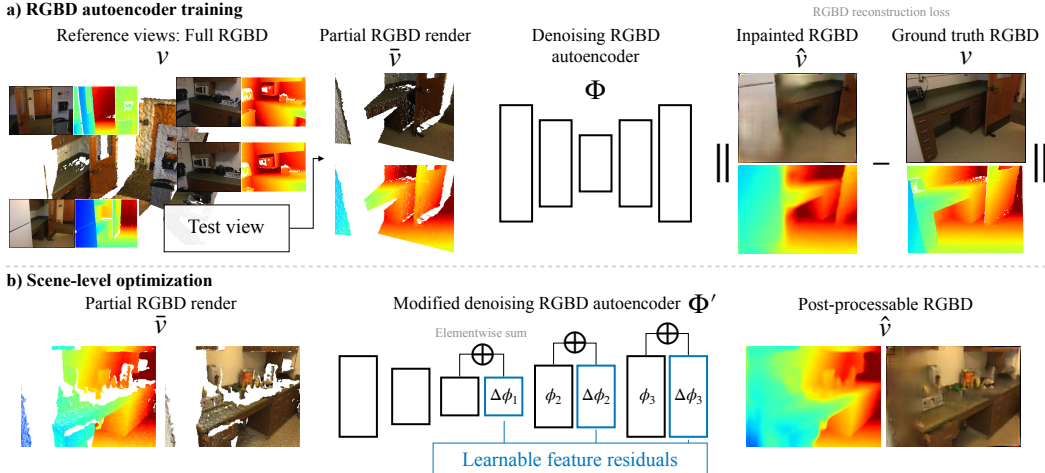

Figure 3: We first train a denoising RGBD autoencoder $\Phi$ for an inpainting task on a large dataset of partial renders of indoor scenes (a). During the scene-level optimization stage (b), $\Phi$ is altered by adding learnable feature residuals $\Delta\phi$ that are summed with the outputs of intermediate layers $\phi$ in order to define a latent parametrization of test scene views.

(DBIR) [42, 3], this approach is not applicable in our case due to the large distances between camera centers that cause occlusions between pixels that DBIR cannot resolve. Instead, we make use of a differentiable *point tracer* [36] that projects the whole scene point cloud $X$ into each of the test views, accounting for occlusions in the process (description of the renderer is deferred to the supplementary). Since the reference views are selected in a sparse manner, the resulting scene point cloud, upon rendering, generates a mere partial render $\bar{v}^i$ in each of the test cameras $(\hat{K}^i, \hat{g}^i)$.

The next step aims to infill the missing parts of the partial renders $\bar{v}^i$. To this end, we train a deep denoising RGBD autoencoder $\Phi(\bar{v}^i) = \hat{v}^i$ which accepts $\bar{v}^i$ and returns a prediction of the full image $\hat{v}^i$ (fig. 3a). $\Phi$ comprises a feature pyramid network (FPN) [25] trunk terminated by a 3x3 convolutional filter with 5 output channels (three RGB channels and additional two for depth and its confidence) and a bilinear upsampler that resizes the output producing a clean RGBD frame $\hat{v}^i$ of the same spatial resolution as $\bar{v}^i$. Training minimizes the inpainting loss from [26] for the RGB channels and the uncertainty-based error defined as a likelihood of predicted parameters of a Laplace distribution over the set of output depth values [30]. Additionally, the network contains two more RGBD prediction branches attached to the output of the 2nd and 3rd upsample&add layer. These predict two more additional inpainted RGBD frames at a lower resolution, which are later passed together with the high-resolution output to the RGBD inpainting losses.

In practice, $\Phi$ is capable of correcting small defects caused by e.g. rendering irregularly sampled surfaces. However, it struggles with larger missing areas where semantically consistent structures, such as pieces of furniture, have to be hallucinated. Indeed this problem is known to be very challenging due to its ambiguous nature where different inpaintings provide a reasonable explanation of the partial render. Instead of sharp predictions, $\Phi$ produces an average over possible solutions manifesting as a blurry RGBD output. Surprisingly, we observed the same behavior for a GAN-based architecture [45] which has been designed for our scenario with a strongly multimodal output space.

The second main failure mode is an inpainting inconsistent across different views of the same underlying 3D surface. This is expected since the denoiser $\Phi$ is applied independently to each partial render $\bar{v}$. A possible solution is to reason in a common 3D space by applying a 3D ConvNet as in [20]. Unfortunately, our experiments revealed that the low resolution of the underlying voxel grid again leads to blurry RGB predictions. The next section describes how we deal with both problems.

## 3.2 Scene-consistent inpainting

In this paper, we propose to tackle the problem of ambiguity and inconsistency with PerspectiveNet, a novel approach that jointly refines the set of test and reference views in order to obtain a globally

consistent solution that respects the geometric constraints of the scene camera system. Performing local scene-specific optimization allows to select one of the possible solutions, resolving the ambiguity issue. To deal with the scene-consistency conundrum, we leverage the depth predicted by the denoiser $\Phi$ and back-project every pixel into the scene point cloud to derive multi-view consistency constraints that guide the image inpainting on a global scene level.

In abstract terms, we pose the scene-centric image inpainting task as a minimization problem of an objective $\mathcal{L}$ of the following form:

$$\mathcal{L} = \min_{\{\hat{\phi}^1, ..., \hat{\phi}^{N_\text{test}}\}} \sum_{i=1}^{N_\text{test}} \ell_\text{cons}\big(\hat{v}^i \,\big|\, V \setminus \{\hat{v}^i\}\big) \quad, \quad \hat{v}^i = \Psi(\hat{\phi}^i) \tag{1}$$

Here, $\ell_\text{cons}(\hat{v}^i | V \setminus \hat{v}^i)$ measures how geometrically consistent an inpainted image $\hat{v}^i$ is with the set $V \setminus \hat{v}^i$ of the other inpainted / reference views in the camera system of the scene. The optimization is over the set of latent representations $\hat{\phi}^i$ of each test view $\hat{v}^i = \Psi(\hat{\phi}^i)$. $\Psi$ is a mapping between a latent space and the space of RGBD images. Intuitively, the minimizer of $\mathcal{L}$ constitutes a globally scene-consistent set of RGBD views $\hat{v}^i$. We now describe two main ingredients of our method: (a) the parametrization function $\Psi$ of the input images, (b) our choice of consistency losses $\ell_\text{cons}$.

**Latent image parametrization** Since optimizing over raw RGBD values is known to be difficult and often requires multiple regularizers, so we know we need a sophisticated function $\Psi$. We follow [3], who demonstrated that a deep latent coding of depth images can overcome the need for complex regularizers. In this work, we opt for a simple solution that leverages the RGBD denoising autoencoder $\Phi$ from section 3.1 above.

We make use of the intermediate feature planes of $\Phi$ to create a latent representation $\phi(\hat{v})$ of each test view $\hat{v}$. Since denoising autoencoders are known to learn generic image representations, optimizing over $\phi(\hat{v})$ is likely to produce a tensor lying on the manifold of plausible RGBD images. This effectively avoids using additional complex regularizers.

More specifically, after training $\Phi$, we convert it into its modifiable version $\Phi'(\bar{v}, \Delta\phi(\bar{v}))$ (illustrated in fig. 3 (b)). The input to $\Phi'$ is a partial render $\bar{v}$ as well as a tuple $\Delta\phi(\bar{v}) = (\Delta\phi_1(\bar{v}), ..., \Delta\phi_L(\bar{v}))$ of feature residuals $\Delta\phi_l(\bar{v})$ that are element-wise added to each of the $L$ intermediate feature tensors produced by the feed-forward pass of $\Phi(\bar{v})$. More formally, $\Phi'$ is defined as:

$$\Phi'(\bar{v}, \Delta\phi(\bar{v})) = \Phi_L(\,...\,\Phi_1(\Phi_0(\bar{v}) + \Delta\phi_1)\,...\,+ \Delta\phi_L), \tag{2}$$

where $\Phi_l$ stands for the $l$-th layer of network $\Phi$. To avoid unnecessary image overparametrization we add the feature residuals $\Delta\phi_l$ only to a preselected subset of feature layers $l$ from the decoding part of $\Phi$. In this manner, $\Phi$ then replaces the latent mapping $\Psi$ in eq. (1), while the tuple $\Delta\phi(\bar{v}^i)$ corresponds to the latent image representation $\hat{\phi}^i$. Having defined a convenient way of parametrizing images in our camera system, next we devise constraints that drive our scene-level inpainting optimization.

**Reprojection consistency loss** Our main constraint ensures that newly generated points in a given inpainted test view $\hat{v}^i$ are consistent with the projection of the scene point cloud $\hat{X}^i$ formed by rendering all other views into $v^i$.

More formally, for each test view $\hat{v}^i$, we form a view-specific point cloud $\hat{X}^i$ by back-projecting into the common scene coordinate frame all pixels from the set $V \setminus \{\hat{v}^i\}$ consisting of all reference and test views excluding $\hat{v}^i$ itself. The point cloud $\hat{X}^i$ is then rendered into camera $(\hat{g}^i, \hat{K}^i)$ forming a *contextual render* $\check{v}^i$. For each test view $\hat{v}^i$ we then define a multiview inpainting consistency loss $\ell_\text{cons}(\hat{v}^i)$ as follows:

$$\ell_\text{cons}\big(\hat{v}^i \,\big|\, V \setminus \{\hat{v}^i\}\big) = \sum_{u \in \Omega(\hat{v}^i) \cap \Omega(\check{v}^i)} h(\hat{v}_u^i, \check{v}_u^i), \qquad h(a,b) = \delta \sum_{\mathbf{c}=1}^{6} \left( \sqrt{1 + \delta^{-1}(a_\mathbf{c} - b_\mathbf{c})^2} - 1 \right),$$

$$\tag{3}$$

which is defined over all pixel locations $u$ that have a non-hole status in both $\hat{v}^i$ and $\check{v}^i$. $h(a,b)$ is an accumulation of Pseudo-Huber losses [5] across dimensions $\mathbf{c}$ of per-pixel RGBXYZ vectors $a, b \in \mathbb{R}^6$. Here, the XYZ component is a backprojection $x_u^i \in \mathbb{R}^3$ of the depth value $d_u$ into the 3D coordinate frame of camera $i$. $h(a,b)$ is further accumulated over 6 scales of a Gaussian image pyramid. We set $\delta = 1$.

**Style consistency loss**  The style loss [14] has been shown to facilitate more realistic results for image generation [6] as well as image inpainting [26] tasks. We adopt the loss in the following form:

$$\ell_{\text{style}}\big(\{\hat{v}^i\}_{i=1}^{N_{\text{test}}} \,\big|\, \{v^i\}_{i=1}^{N_{\text{ref}}}\big) = \sum_{l \in 1,2,3} \left| \sum_{i=1}^{N_{\text{ref}}} \frac{\Psi_l(v^i)\Psi_l(v^i)^T}{N_{\text{ref}}H_lW_l} - \sum_{i=1}^{N_{\text{test}}} \frac{\Psi_l(\hat{v}^i)\Psi_l(\hat{v}^i)^T}{N_{\text{test}}H_lW_l} \right|_1, \qquad (4)$$

where $\Psi_l(v) \in \mathbb{R}^{D \times H_lW_l}$ denotes a set of features from $l$-th intermediate layer of an ImageNet pre-trained VGG16 network [32] reshaped into a $D \times H_lW_l$ matrix after flattening the last two dimensions of the feature tensor of the original size $D \times H_l \times W_l$. As in [27], we use features extracted after each of the first 3 convolutional layers of VGG16.

Intuitively, the loss pools a style descriptor from reference images and ensures that the newly inpainted pixels match this distribution. Since the style transfer loss is known to produce fish-scale artifacts, following [26], we use it in conjunction with a total variation regularizer $\ell_{\text{TV}} = \frac{1}{N_{\text{test}}WH} \sum_{i=1}^{N_{\text{test}}} \sum_{u_1,u_2} |c^i_{(u_1,u_2+1)} - c^i_{(u_1,u_2)}| + |c^i_{(u_1+1,u_2)} - c^i_{(u_1,u_2)}|$, where $c^i_{(u_1,u_2)}$ is the RGB value of a pixel at position $(u_1, u_2)$ in a test view $\hat{v}^i$.

**Scene-level optimization**  Having defined the main constraints and image parametrization, we can now write the objective $\mathcal{L}$ of the PerspectiveNet scene-level optimization:

$$\mathcal{L} = \underset{\{\Delta\phi(\bar{v}^i)\}_{i=1}^{N_{\text{test}}}}{\arg\min} \ \ell_{\text{style}}\big(\{\hat{v}^i\}_{i=1}^{N_{\text{test}}} \,\big|\, \{v^i\}_{i=1}^{N_{\text{ref}}}\big) + \sum_{i=1}^{N_{\text{test}}} \ell_{\text{cons}}\big(\hat{v}^i \,\big|\, V \setminus \{\hat{v}^i\}\big) \qquad \hat{v}^i = \Phi'(\bar{v}^i, \Delta\phi(\bar{v}^i))$$
(5)

For a given scene, $\mathcal{L}$ is minimized with an Adam [21] optimizer for 50 iterations with an initial learning rate of 0.01 decaying 10-fold after 35 iterations.

**PerspectiveNet in a nutshell**  Our algorithm thus works as follows. For each testing scene, the set of reference views $\{v^i\}$ is rendered into the target cameras $\{\hat{g}^j\}$ producing partial renders $\{\bar{v}^j\}$. The ensuing scene-consistent optimization process, that minimizes $\mathcal{L}$ (eq. 5), then finds the optimal set of latent image representations $\{\Delta\phi(\bar{v}^i)\}$ that, after being passed with $\{\bar{v}^j\}$ to the modifiable denoising autoencoder $\Phi'$, leads to the final set of scene-consistent new views $\{\hat{v}^j = \Phi'(\bar{v}^j, \Delta\phi(\bar{v}^j))\}$.

### 3.3 Technical details

**Additional regularizers**  While $\ell_{\text{cons}}$ ensures that newly generated pixels stay consistent with all views in the scene, in principle, there is nothing stopping the the global optimization process from producing a set of consistent inpaintings that get "detached" from the reference views. We thus add two regularization terms that prevent the solution from diverging too far from the ground truth provided by the reference frames:

$$\ell_R = \sum_{i=1}^{N_{\text{test}}} \left[ \sum_{u \in \Omega(\bar{v}_u^i)} h(\bar{v}_u^i, \hat{v}_u^i) + \sum_{u \in \Omega(\hat{v}_u^i)} h(\hat{v}_{u,t=0}^i, \hat{v}_u^i) \right], \qquad (6)$$

where the first term of the main sum brings the non-holes of the partial ground truth render $\bar{v}_u^i$ close to the corresponding pixels in the inpainted image $\hat{v}_u^i$, and the second term prevents the result of the optimization $\hat{v}^i$ from grossly differing from the initial inpainting $\hat{v}_{u,t=0}^i$ obtained by $\Phi$ at the beginning of the global optimization.

**Training the RGBD denoiser $\Phi$**  In order to train $\Phi$, we collect a dataset of image pairs $\{(\bar{v}^i, v^i)\}$ generated by randomly sampling 8 reference views from the training scenes of a considered dataset of indoor scenes and rendering those using our point tracer into a random test view, for which the ground truth RGBD frame $v^i$ is known. We further filter out the pairs where less than 50% of the input pixels are defined. The RGBD autoencoder $\Phi$ is trained with an initial learning rate $10^{-5}$ decaying 10-fold once the loss plateaus. Where possible, the convolutional layers were initialized with weights of an ImageNet pre-trained ResNet-50 network. Batch size was set to 4 and training on a single GPU took approximately 7 days. For each of the 2 datasets considered in this paper (ScanNet [8], SceneNet [28]), we train a separate autoencoder.

(a) **ScanNet**

| Method | Color metrics | | | Depth metrics | | | |
|---|---|---|---|---|---|---|---|
| | $\ell_1^{RGB} \downarrow$ | PSNR $\uparrow$ | LPIPS $\downarrow$ | $\ell_1^D[m] \downarrow$ | $\delta_1 \uparrow$ | $\delta_2 \uparrow$ | $\delta_3 \uparrow$ |
| PerspectiveNet | 68.022 | 13.762 | **0.422** | **0.115** | **0.352** | **0.411** | **0.471** |
| PerspectiveNet w/o opt | **66.511** | **13.986** | 0.426 | 0.120 | 0.188 | 0.230 | 0.279 |
| PartialConv [26] | 93.604 | 11.374 | 0.461 | 0.750 | 0.194 | 0.236 | 0.283 |
| 3DConvNet | 78.590 | 12.190 | 0.531 | 0.138 | 0.301 | 0.359 | 0.426 |
| BiGAN [45] | 77.313 | 12.742 | 0.523 | 0.215 | 0.169 | 0.212 | 0.265 |

(b) **SceneNet**

| Method | $\ell_1^{RGB} \downarrow$ | PSNR $\uparrow$ | LPIPS $\downarrow$ | $\ell_1^D[m] \downarrow$ | $\delta_1 \uparrow$ | $\delta_2 \uparrow$ | $\delta_3 \uparrow$ |
|---|---|---|---|---|---|---|---|
| PerspectiveNet | 49.698 | 15.687 | **0.424** | **0.219** | **0.366** | **0.431** | **0.494** |
| PerspectiveNet w/o opt | **48.521** | **16.324** | 0.442 | 0.227 | 0.101 | 0.125 | 0.155 |
| PartialConv [26] | 76.470 | 12.377 | 0.481 | 1.846 | 0.008 | 0.010 | 0.013 |
| 3DConvNet | 75.942 | 12.614 | 0.570 | 0.653 | 0.040 | 0.050 | 0.062 |
| BiGAN [45] | 55.815 | 15.112 | 0.485 | 0.249 | 0.319 | 0.375 | 0.431 |

Table 1: **Quantitative evaluation of depth and image generation** on the test sets of ScanNet (a) and SceneNet (b) comparing our method with 2D and 3D inpainting baselines.

# 4 Experiments

**Datasets and evaluation protocol**   We chose 2 datasets for evaluation: ScanNet [8] and SceneNet [28]. ScanNet currently comprises one of the largest 3D datasets of real indoor scenes with 1500 training and 100 test scenes. Contrasted to the realistic ScanNet, SceneNet is a dataset of 33k/1k synthetic train/test scenes. SceneNet was chosen in order to benchmark the performance in a clean setting free of challenging factors such as lighting changes or inaccurate camera extrinsics.

Each dataset contains RGBD views of indoor scenes annotated with camera extrinsic and intrinsic parameters, allowing for evaluation of the new view synthesis. In order to benchmark a method on a given test scene, we sample 4 reference views, for which we assume knowledge of their RGBD as well as camera parameters, and at most 8 reference views for which only the camera parameters are given. For the test views, we then generate the color and depth channels and compare them to the corresponding ground truth frames. In order to obtain good coverage of the scene contents, the reference views were selected by clustering the camera pose descriptors (consisting of a concatenation of the vectorized camera rotation matrix and the camera translation vector) into four clusters and picking the typical point from each cluster as a reference camera. The test views were chosen in a similar fashion by clustering the parameters of the remaining cameras and picking the views that contain at least 50%/40% defined pixels for ScanNet and SceneNet respectively after rendering the contents of the reference views. For each dataset, we first train all methods on the frames coming from its training scenes. For ScanNet, the evaluation is conducted on all 100 test scenes, and for SceneNet we randomly sampled 100 scenes from the test set for evaluation. We produce images of width/height 320/240 pixels and compare with the ground truth images at the resolution of 640/480.

Following standard practice [45, 38, 26, 40, 18], we quantitatively evaluate generated images by reporting the per-pixel $\ell_1$ error ($\ell_1^{RGB}$) and the peak-signal-to-noise-ratio (PSNR). Since $\ell_1$ loss and PSNR are known to be overly sensitive to errors in low-level image details while being insensitive to more abstract semantic visual structures, we also evaluate the perceptive error LPIPS [41], which is a calibrated version of a distance between images in a feature space of a pre-trained image classification network (VGG16 in our case). In order to evaluate the generated depth maps, following [24, 11], we report per-pixel absolute depth error ($\ell_1^D$ measured in meters) and metrics $\delta_i$ for $i = \{1, 2, 3\}$ which measure the portion of test pixels that have their absolute depth error lower than a threshold $t_i = 1.25^i$ cm.

**Inpainting baselines**   Evaluation focuses mainly on inpainting baselines that consist of rendering the reference views into the target ones, followed by filling the holes with an algorithm. The baseline abbreviated as **PartialConv** uses a state-of-the-art inpainting architecture from [26] trained on the same dataset as our RGBD denoiser. We also compare with **BiGAN** [45] trained on the same dataset. Finally, **PerspectiveNet w/o opt** is an ablation of our method and comprises the initial

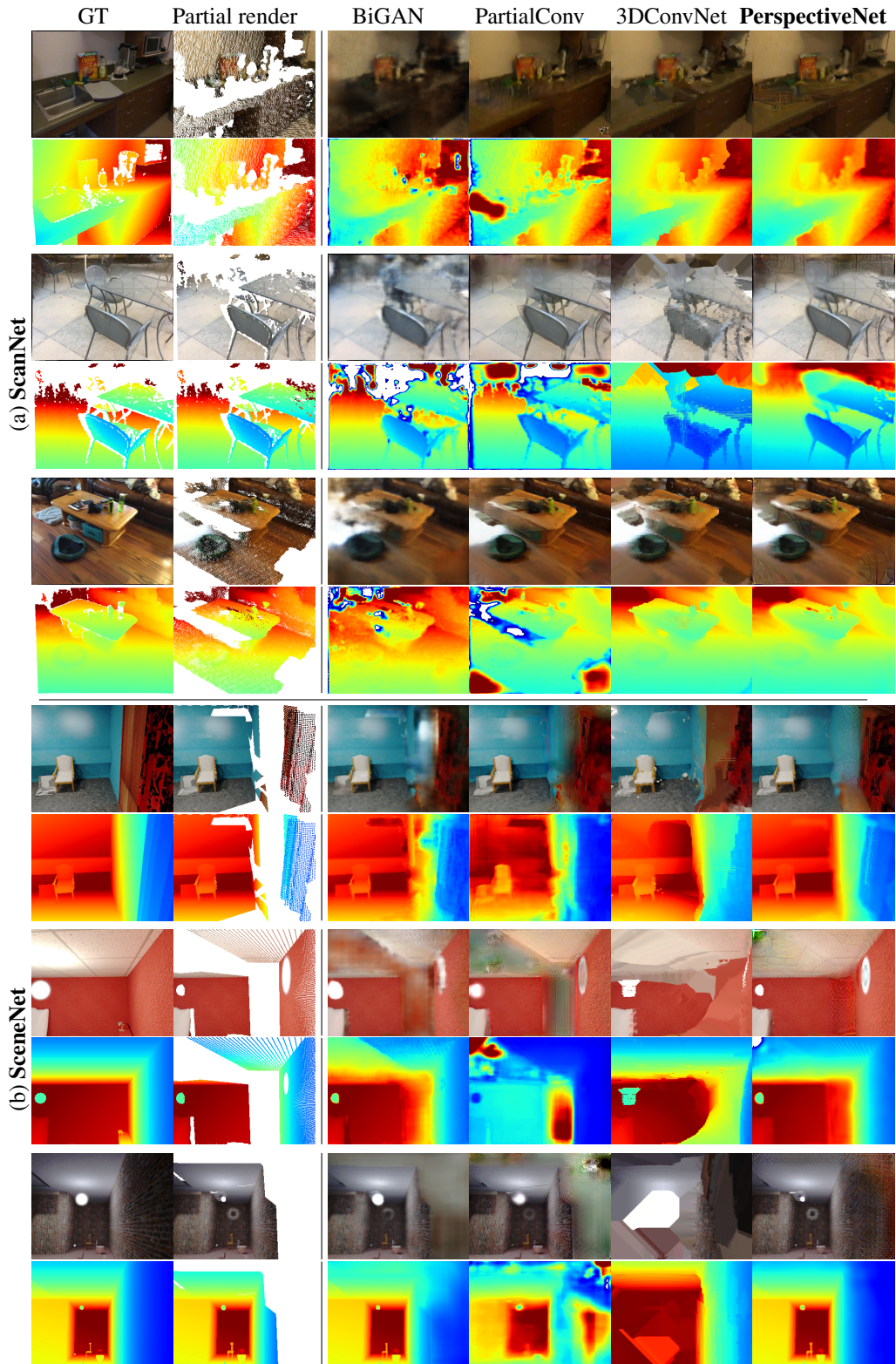

Figure 4: **Qualitative evaluation of new RGBD view synthesis on the ScanNet (a) and SceneNet (b) datasets** comparing our method (PerspectiveNet) to inpainting with partial convolutions (Partial-Conv [26]) or Bicycle GAN (BiGAN [45]), and a sparse 3DConvNet that inpaints voxels directly in 3D. The first column of each row denotes the ground truth for a given test view while the second shows a partial render $\bar{v}$ of the reference views into the camera of the ground truth view. For each of the 6 displayed test cases, we show the RGB (upper row) and depth prediction (lower row).

inpainting produced by the RGBD denoiser from section 3.1 without any iterations performed by the scene-consistent optimizer.

**3D inpainting**  Apart from the inpainting methods, we further compare with an approach that operates on 3D voxel grids (**3DConvNet**). Since our application requires a voxel grid of a very high resolution and spatial extent, due to high memory requirements of the classic dense 3D ConvNet architectures, we implemented a sparse U-Net convolutional network [15]. Detailed explanation of the architecture is included in the supplementary material.

Table 1 contains quantitative results on the ScanNet and SceneNet datasets. A qualitative comparison is present in fig. 4. Additional qualitative results are present in the supplementary material.

**Discussion of results**  Table 1 reveals that our method outperforms the considered baselines on all depth metrics. For the color metrics $\ell_1^{RGB}$ and PSNR, we are on par with the ablation "PerspectiveNet w/o opt". However, we outperform it on the more semantically meaningful LPIPS metric. Intuitively, since PSNR and $\ell_1^{RGB}$ are sensitive to low-frequency image details and LPIPS better assesses image realism, the relative differences in the color metrics between PerspectiveNet and "PerspectiveNet w/o opt" signify that, while the local color distributions are roughly correct in both cases, adding the scene-consistent optimizer brings better image realism.

Qualitatively, compared to our approach, the inpainting baseline PartialConv generates more blurry results due to a suboptimal loss function that does not take into account the ambiguity in the output. Furthermore, PartialConv records low performance of depth inpainting. Our method also outperforms BiGAN. For BiGAN, we observed that changes in the latent code $z$ mostly lead to global change of the color statistics of the output image, rather than altering the geometry of the inpainted scene. 3DConvNet records a competitive depth prediction accuracy, but lags behind in color prediction. This is most likely due to the reconstructions being optimized to match the partial point clouds in 3D, without considering the need for perceptual realism when rendering the voxels into the 2D test views.

## 5  Conclusion

In this work, we tackled a previously seldom explored problem of new-view synthesis in real indoor environments. A novel approach, termed PerspectiveNet, based on the render-inpaint paradigm is proposed. The main technical contribution is a bundle-adjustment technique that jointly optimizes all views in a given room in order to obtain a set of new views that is globally scene-consistent in terms of geometry and style. Evaluation on two large datasets of indoor scenes [8, 28] reveals performance superior to several strong baselines.

## Footnotes

[1]We use upper indices to index frames, while lower indices stand for spatial locations of pixels within a frame

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
