[Supplementary Material]

# PerspectiveNet: A Scene-consistent Image Generator for New View Synthesis in Real Indoor Environments

## *Supplementary material*

## 1 Additional training details

**Learning details of the RGBD denoising autoencoder** $\Phi$ The RGBD autoencoder $\Phi$ is trained on the dataset of partial renders from ScanNet for 150 epochs using Adam optimizer with initial learning rate $10^-5$ decaying 10-fold after visiting 100k minibatches. Where possible, the convolutional layers were initialized with weights of an ImageNet pre-trained ResNet50 network. Batch size was set to 10 and training on a single GPU took approximately 4 days.

**Loss weighting** The individual terms of the overall loss $\mathcal{L}$ were weighted as follows. $w(\ell_{\mathrm{TV}}) = 0.01$, $w(\ell_{cons}) = 0.01$, $w(\ell_{style}) = 0.01$. For $w(\ell_R)$, the first term of the outer sum was weighted with $w(\ell_R^1) = 1$ and the second with $w(\ell_R^2) = 0.1$. The weights were optimized by conducting a grid search on a subset of the training set. For each hyperparameter, the range of tested values was (1.,0.1,0.01).

## 2 Differentiable bi-linear splat rendering

This section describes our differentiable point tracer used for rendering scene point clouds to create partial renders $\hat{v}$ and also to derive reprojection consistency losses.

In order to render a point cloud $X = \{x_j\}_{j=1}^M$, each point $x_j$ is first projected into the camera frame with $[u_{1j} \ u_{2j} \ d_j \ 1]^T \sim Kg[x_j^T \ 1]^T$. Using those projected floating-point coordinates $(u_{1j}, u_{2j}, d_j)_{m=1}^m$ with $d_i > 0$ and their color vectors $c_j \in [0,1]^3$, for each $(a,b) \in \mathbb{Z}^2$, we calculate a weight $w_{a,b,j} = (1 - |u_{1j} - a|)(1 - |u_{2j} - b|)\exp(-d_j)$ if $|u_{1j} - a| < 1$ and $|u_{2j} - b| < 1$, and zero otherwise. Each cell $(a,b)$ in the lattice of the render is then colored with a weighted sum of the $c_j$ with normalized weights $w_{a,b,j}/\sum_{j'} w_{a,b,j'}$.

## 3 Detailed explanation of 3DConvNet

In what follows, we provide additional details about the 3D voxel-grid based architecture 3DConvNet benchmarked in the experimental section.

The receptive field of the SparseConvNet U-Net is a $512^3$ grid of 2cm-cubed, RGB-valued voxels. This corresponds to a physical region of size appromixately 10m-cubed. During training, the input is a sparse point cloud derived from the 4 reference views. The network is trained to predict the dataset's 'ground-truth' point cloud that was obtained using all the views of that scene.

The 'left' half of the U-Net downsamples the input to size $256^3$, then $128^3$ and so on using a mixture of submanifold sparse convolutions and sparse strided-convolutions. The 'right' half of the U-Net upsamples the hidden states from $1^3$ back up $512^3$ using strided transpose convolution. After each transpose convolution the network predicts which of the new voxels should be occupied. These predictions are used at test time to predict a full-scene point cloud. Shortcut connections pass information from left to right at the same spatial scale whenever the same location is active on both sides.

On the test set, we post-process the predicted point cloud by:

- Removing any voxels that cannot be present based on the empty space that can be inferred to exist inbetween the context camera positions and the corresponding observed points in the input partial point cloud.
- Setting the RGB values of voxels that are present in the input to that value.
- Setting the values of the new voxels iteratively by propagating the values from known neighbors.

We then project the result to the test views and calculate the aforementioned metrics.