[Reviews · NeurIPS 2019]

Reviewer 1



Given few RGBD images of a real indoor scene as well as camera locations where these were taken, the algorithm predicts RGBD images takes from different camera locations. The novelty is the use of denoising auto-encoder for a given view and finding latent representations that are consistent for different views. Detailed comments: - It would be good if the whole process was described in steps because it wasn’t clear what the overall approach is from the start (may be it would be for someone working on a similar topic). Some figures are good, but could be better - together with such description. Something like the following would be useful for me: A) We are given a set of RGBD views along with camera locations of a given scene. Given a subset of these views we would like to predict the other set (given camera locations) B) For each given view we can map each RGBD pixel into 3d space using plain geometry. C) We can then project these pixels into a new view again using plain geometry. Some of the pixel locations in the new view will not be filled, we called those holes. Some of them will have multiple guess - we use a heuristic that choses these values as … (explain in one sentence) D) For each view we train a denoising auto-encoder to fill missing details of a given view. E) To make the filled details self consistent we do the following. For each view we modify the latent to be consistent with other views as follows: We take latent of n-1 views, project them to pixels, then extract the 3d locations of these pixels, project them to the remaining view and impose a loss that makes this view similar to that decoded from the corresponding latent representation of that view. We optimise all the latents by gradient descent. There could be a figure going with that ,or perhaps what you already have but with this kind of description in the caption. 86: It would be good to describe g and K of the camera. 94: …describe what are non-holes 102: May be renderer shouldn’t be in the supplement, seems like the fundamental component. May be at least explain intuitively what it is doing (especially for people not familiar with this literature). Why exp(-d) (from supplement) - so its differentiable? Gan: It is likely that you gan is not well trained and could be made better. It would also be good to train both - simple reconstruction to ground it and create a good representations and then gan to hallucinate the right details. - It would be good if a simpler solution was found - there are lot of regularisers and choices here. How much hyper parameter optimisation was there. - Show more source, target and unpainted images (in supplement say) - just to see more results of how well it work. 167: Why only non-holes of the second v since the first one is auto encoded and has values everywhere. Pros: Nice use of denoising auto encoder and the self-consistency training Cons: - Relying on a depth Channel - which provide a clear grounding of each pixel and mapping between different views - it would be much better if such can things can be inferred. - Blurry filling - using a proper generative model would be good (non-gan might work for this as well)

Reviewer 2



Main reason for "accept" decision - addressing the problem on realistic, large scale indoor scene dataset; nice theoretical contribution on the losses, and explaining the decisions made. Good points: - tackles novel, challenging, large scale problem of synthesizing views for indoor scenes; - works on a large dataset of realistic indoor scenes. - introduces reprojection consistency loss and style consistency loss, which is a nice theoretical contribution - this work is relevant in the context of indoor localization and navigation applications, where inpainting is necessary, e.g. for completing meshes, reconstructing views for which information is not available, such as holes due to occluders. Not so good points: - Evaluates only on one dataset, while other (larger) scale indoor scene dataset exists (Matterport3D, Gibson). If not suitable, explain why. Abstract and figure 1 -- omission of the mention that the input should be RGBD. It is not clear from the evaluation how the method performs on synthetic, smaller scenes. Abstract: mention that as few as 4 RGB”D” views are needed. The mention of reference views taken with a hand-held camera is a bit misleading, making the readers expect just RGB. Figure 1: mention that the input should contain depth as well. The reference input views are captured at the same time as the desired output views. Might be a good idea to emphasize that as few as 4 reference views In related work (L72), GQN has not been tested in real world setup -- it would have been valuable to add this experiment for comparison. Since the method is capable of browsing simple synthetic experiments, it would be worth to check how it performs on realistic Figure 1 and 2 are not referenced in the text. It is unclear later on whether the 3D Conv Net[34] (L72) mentioned in table 1 and later L234-240 is a contribution of the current work or it was proposed in [34]. 3. L90-91: what is the range for d_u (depth) ? what are the units? L102 -- the differentiable point tracer is suitable to be a part of the main paper. Is it mandatory or important for the point tracer to be differentiable? 3.2 L154-158: how are the layers pre-selected for adding residuals? 4. Experiments L206: -- Matterport3D (https://arxiv.org/pdf/1709.06158.pdf) is larger indoor scene dataset; they provide, among others, RGBD + camera annotations. How would the proposed method perform on MP3D? (Optional) evaluate on Gibson dataset (http://gibsonenv.stanford.edu/database/). L209, 210 → are the 8 views used for testing, i.e., reference views L213: how are the views clustered? Is this a manual step? if not, what features were used? How would the current method perform on ShapeNet? How difficult would it be to compare with methods that evaluate on ShapeNet, e.g. Dosovitskyi et al → how would it perform on SceneNet? There is no clear comparison showing why the proposed method For ablation: what is the contribution of individual loss components? e.g. style consistency vs reprojection consistency? Bring the dataset statistics closer to the beginning of the section defining the evaluation protocol (# of samples, #train, #test); It is not clear how the train / test were split, and whether there is a validation set. In Figure 4 - please add the color and depth measures for the selected pictures (e.g. similar to Table 1). In discussion -- It is understandable from the text and the table 1 that the authors are comparing against a very strong baseline (ablation). How much does 0.02 in PSNR (color), or 0.03 in LPIPS affect perception? For someone not familiar with these measures, how could one understand the improvement? The meaning of these differences should be explained in the discussion of the results.

Reviewer 3



> The BiGAN image predictions seem noisy along the grid. This might be the result of suboptimal architectural choices (low model capacity, filter size etc). > This approach seems quite similar to "Neural Rerendering in the Wild" (Meshry et al) at CVPR 2019. This paper uses a similar approach of using point cloud representations in the context of multi-view reconstruction. How are these methods related? > It would be good to get clarity on how different this work is from Meshry et al. before evaluating contributions in this paper. I hope that the rebuttal clarifies this.

[Author Response · NeurIPS 2019]

We thank reviewers for their insightful comments. Please find below our answers to the questions.

**R1: Describe PerspectiveNet in more clear steps. Describe $g$, $K$, non-holes.** Thank you, we will add a clear
overview of the algorithm as suggested and expand ln.86-94 with more concrete descriptions.

**R1 & R2: Move point-tracer from supplementary.** We agree and we will migrate the paragraph to the paper.
Weighting with $\exp(-d)$ enables differentiability which is a mandatory requirement for optimizing $\ell_{cons}$.

**R1 & R3: BiGAN predictions noisy. Incorrectly trained?** During preliminary experiments, we observed "red flags"
related to GANs, suggesting autoencoders are more suitable: (1) Training a state-of-the-art MSGGAN [Karnewar et al.:
MSG-GAN ...] on SceneNet lead to unrealistic blurry results (fig. I). (2) Insufficient coverage of the image distribution,
whose evidence was an inability to recover latent codes that lead to a correct reconstruction of arbitrary held-out images.

**R1: Hyperparam opt?** Grid search over 3 weights $\{10^{-i}\}_{i=0}^{2}$ for each of 3 losses on 100-scene subset of the train set.

**R1: Show more images.** As suggested, we will expand the supplementary with more qualitative results.

**R1: Why optimizing only non-holes of $\check{v}$ (Eq. 3)?** $\check{v}$ is a point cloud render and can contain holes. Minimizing
$h(\hat{v}_{\bar{u}}, \check{v}_{\bar{u}})$ over holes $\bar{u}$ would make $\hat{v}_{\bar{u}}$ attain an unrealistic color of a hole (black by default) which is not desireable.

**R1: Blurry filling. Use GAN?** Unfortunately, as mentioned above, training a GAN lead to unrealistic blurry results.

**R1: The approach requires depth as input.** We have now imple-
mented a method requiring ground truth (GT) depth solely at train
time. We replaced the GT reference view depth with an output of a
depth predictor [Laina et al.: Deeper ...] trained on the ScanNet train

set. Again, PerspectiveNet outperforms other baselines (Table Ia).      Figure I: ScanNet-trained MSGGAN samples.

**R2: Not mentioning depth as a required input.** We will update the text accordingly to avoid misleading readers.

**R2: Evaluation only on 1 dataset.** As suggested, we have now conducted evaluation on Matterport3D and SceneNet
(same train/test protocol as for ScanNet). Note that SceneNet is synthetic and composed of ShapeNet objects and,
hence, is more suitable for our scene-centric setting than the object-centric ShapeNet. Tables (Ib) and (Ic) contain
results of our experiments. Similar to Tab. 1 in paper, PerspectiveNet outperforms other approaches. Unfortunately, due
to limited amount of time, we could not finish all 3DConvNet experiments (we will include them in camera-ready).

**R2: Test GQN on real data?** We have now trained&tested GQN on ScanNet. GQN failed to learn and attained poor
quantitative results - Ours/GQN: $\ell_1^{RGB}$=67.77/165.70, PSNR=13.79/6.96, LPIPS=0.434/0.687, $\ell_1^{D}$=0.109/0.513. The
failure to learn probably occurs due to a greater complexity of ScanNet compared to GQNs' simplified synthetic scenes.

**R2: Is 3D ConvNet a contribution?** The 3D ConvNet was designed as a baseline we compare with.

**R2: Range/units of depth $d_u$?.** The depth is always expressed in meters. Range is rougly $[0.2, 7]$ meters.

**R2: Which layers for residuals?** $\Delta\phi^i$ were added after every "upsample&add" layer of FPN (four $\Delta\phi^i$ in total).

**R2: L209: Are the 8 views used for testing?** The 4 reference views provide all geometry and appearance conditioning.
Hence, inpainting and evaluation happens only for 8 test views, for which we only know the camera parameters.

**R2: View clustering?** Given $N$ cameras, we KMeans-clustered the set of corresponding descriptors $\{\text{vec}(g^i)\}_{i=1}^{N}$.

**R2: Loss weights? Train/test split?** $w(\ell_{style}, \ell_{cons}, \ell_R) = (0.1, 0.01, 0.1)$. Using official train/test split of ScanNet.

**R2: Explain perception of improvements in LPIPS / PSNR / l1.** PSNR and $\ell_1^{RGB}$ are sensitive to low-frequency
image details while LPIPS better assesses image realism. Hence, the +8/-1% improvement of *PerspNet* over *PerspNet*
*w.o. opt* in LPIPS / PSNR means that, while the local color distributions are roughly correct in both cases, adding the
scene-consistent optimizer brings better image realism and an image-to-image consistent inpainting.

**R2: Performance analysis.** While PerspectiveNet brings better image quality, it is fair to admit that this comes at the
cost of sub-real-time execution times (~20s per scene).

**R3: Discuss differences with [Meshry et al.].** We agree that there are similarities with the work of Meshry et al.
[a] and we will cite this paper in Sec. 2. However, *our work differs substantially in*: (1) The task: While we focus
on precise reconstruction of geometry and appearance of a scene given a limited amount of information in form of
an image with large undefined regions, [a] is a form of stylization that aims at capturing a complete distribution of
possible appearance variations of a, mostly hole-free, image. (2) Available data: [a] uses 1000s of reference images to
reconstruct a scene that is later re-rendered. We use only 4 reference views, leading to large holes in new views and
significantly harder inpainting problem. Furthermore, [a] requires semantic segmentation of the scene.

Finally, please note that [a] uses a BiGAN approach which we compare with in our work and outperform it significantly.

| Dataset | (a) ScanNet w/o test-time GT depth | | | | (b) SceneNet | | | | (c) Matterport3D | | | |
|---|---|---|---|---|---|---|---|---|---|---|---|---|
| Metric | $\ell_1^{RGB} \downarrow$ | PSNR $\uparrow$ | LPIPS $\downarrow$ | $\ell_1^{D} \downarrow$ | $\ell_1^{RGB} \downarrow$ | PSNR $\uparrow$ | LPIPS $\downarrow$ | $\ell_1^{D} \downarrow$ | $\ell_1^{RGB} \downarrow$ | PSNR $\uparrow$ | LPIPS $\downarrow$ | $\ell_1^{D} \downarrow$ |
| PerspectiveNet | **93.819** | 11.193 | **0.515** | **0.505** | **51.722** | **15.442** | **0.521** | **0.214** | **38.905** | **19.108** | **0.404** | **0.226** |
| PerspectiveNet w/o opt | 94.333 | **11.224** | 0.537 | 0.516 | 61.493 | 14.950 | 0.564 | 0.280 | 42.173 | 17.722 | 0.457 | 0.384 |
| PartialConv | 96.742 | 10.948 | 0.515 | 0.606 | 80.612 | 12.218 | 0.545 | 1.984 | 46.741 | 17.119 | 0.411 | 0.647 |
| 3DConvNet | - | - | - | - | 75.942 | 12.614 | 0.614 | 0.653 | - | - | - | - |
| BiGAN | 156.958 | 7.194 | 0.715 | 0.666 | 99.358 | 11.106 | 0.637 | 0.841 | 118.614 | 9.940 | 0.613 | 1.286 |

Table I: Additional results on test sets of Matterport3D, SceneNet, ScanNet (will be included in camera-ready).

[Meta-Review · NeurIPS 2019]

This submission received borderline reviews. After the post-rebuttal discussion, the most negative reviewer (who also provided a very cursory review) has come around slightly and is willing to increase their score to a borderline accept (though they have not actually done so in CMT). One of the main concerns was similarity to a piece of prior work (Meshry et al.) However, the rebuttal seems to have sufficiently addressed this--though the methodology is similar, the application is significantly different. Another concern is that the performance of the BiGAN appears quite poor. None of the reviewers seem to have enough expertise to judge whether the model just wasn't trained appropriately (as these models are very tricky to train even for experts), or whether this is a more fundamental problem. Nevertheless, the paper's main contributions don't seem to depend critically on this, so it seems not to be a clear reason to reject the paper.